# Utilizing Dynamic Contrast-Enhanced Magnetic Resonance Imaging (DCE-MRI) to Analyze Interstitial Fluid Flow and Transport in Glioblastoma and the Surrounding Parenchyma in Human Patients

**DOI:** 10.3390/pharmaceutics13020212

**Published:** 2021-02-04

**Authors:** Krishnashis Chatterjee, Naciye Atay, Daniel Abler, Saloni Bhargava, Prativa Sahoo, Russell C. Rockne, Jennifer M. Munson

**Affiliations:** 1Department of Biomedical Engineering & Mechanics, Fralin Biomedical Research Institute, Virginia Tech, Roanoke, VA 24016, USA; krisc83@vt.edu (K.C.); naciye@vt.edu (N.A.); salonib@vt.edu (S.B.); 2Department of Computational and Quantitative Medicine, Division of Mathematical Oncology, Beckman Research Institute, City of Hope, Duarte, CA 91010, USA; daniel.abler@artorg.unibe.ch (D.A.); iitkprativa@gmail.com (P.S.); rrockne@coh.org (R.C.R.); 3ARTORG Center for Biomedical Engineering Research, University of Bern, 3008 Bern, Switzerland

**Keywords:** glioblastoma (GBM), DCE-MRI, interstitial flow, convection, diffusion, Cancer Imaging Archive

## Abstract

Background: Glioblastoma (GBM) is the deadliest and most common brain tumor in adults, with poor survival and response to aggressive therapy. Limited access of drugs to tumor cells is one reason for such grim clinical outcomes. A driving force for therapeutic delivery is interstitial fluid flow (IFF), both within the tumor and in the surrounding brain parenchyma. However, convective and diffusive transport mechanisms are understudied. In this study, we examined the application of a novel image analysis method to measure fluid flow and diffusion in GBM patients. Methods: Here, we applied an imaging methodology that had been previously tested and validated in vitro, in silico, and in preclinical models of disease to archival patient data from the Ivy Glioblastoma Atlas Project (GAP) dataset. The analysis required the use of dynamic contrast-enhanced magnetic resonance imaging (DCE-MRI), which is readily available in the database. The analysis results, which consisted of IFF flow velocity and diffusion coefficients, were then compared to patient outcomes such as survival. Results: We characterized IFF and diffusion patterns in patients. We found strong correlations between flow rates measured within tumors and in the surrounding parenchymal space, where we hypothesized that velocities would be higher. Analyzing overall magnitudes indicated a significant correlation with both age and survival in this patient cohort. Additionally, we found that neither tumor size nor resection significantly altered the velocity magnitude. Lastly, we mapped the flow pathways in patient tumors and found a variability in the degree of directionality that we hypothesize may lead to information concerning treatment, invasive spread, and progression in future studies. Conclusions: An analysis of standard DCE-MRI in patients with GBM offers more information regarding IFF and transport within and around the tumor, shows that IFF is still detected post-resection, and indicates that velocity magnitudes correlate with patient prognosis.

## 1. Introduction

Glioblastoma (GBM) is the most lethal form of brain tumor, with a median lifespan post-diagnosis of 12–15 months and a 100% recurrence rate, often within several centimeters of the resection cavity. The most recent advancement in GBM therapy was the implementation of concurrent radiotherapy and temozolomide reported by Stupp et al. in 2005 [1]. This study pioneered the current standard of care for primary GBM as complete resection, if possible, followed by concurrent temozolomide and radiotherapy administration. Unfortunately, the Stupp protocol only led to a slight improvement in outcomes for patients, increasing the overall five-year survival to only 27.2% for primary GBM. Despite extensive research efforts since, there has been no advancement in the overall survival of GBM. Though preclinical models have exhibited efficacy of several treatment options and therapeutic agents, this success has not translated to clinical trials [2]. A major impediment to the translation of treatments from the bench to the bedside is the inability to effectively deliver therapeutics within and around the tumor [3].

There are two main modes of drug delivery used in a clinic—systemic and local. Though systemic delivery is often less invasive and easier to implement, it must pass through several hurdles before possibly resulting in a therapeutically effective response. These challenges include a risk of drug degradation and clearance by the rest of the body, intolerable systemic toxicity, and inability to cross the blood–brain barrier (BBB) at the tumor site [4]. The fact that GBM is a vascularized tumor with abnormal, leaky neovasculature (often with a disrupted BBB) can be advantageous for systemic drug delivery [5,6]. In fact, this leaky vasculature is exploited to visualize GBM using a combination of paramagnetic contrast agents such as gadolinium and magnetic resonance imaging (MRI). However, despite the presence of areas with a disrupted BBB, the areas of the adjacent brain vasculature with an intact BBB are enough to limit the delivery of drugs to invading tumor cells [7]. Additionally, the high pressures that contribute to a blood–tumor barrier (BTB) and reduce the transport of drugs from the vasculature into the interstitial space, thus reducing therapeutic delivery [8,9]. The shortcomings of systemic drug delivery prompted the development of local delivery techniques to bypass the BBB and BTB entirely. Types of local drug delivery include the perioperative implant delivery into the resection cavity, intraventricular or intrathecal delivery, and convection-enhanced delivery (CED), all of which aim for therapeutic delivery directly into the tumor and surrounding brain parenchyma [10,11,12].

We believe that the success of both intraventricular or intrathecal delivery and CED is highly dependent on understanding interstitial fluid flow (IFF), which is defined as the flow found throughout both healthy and peritumoral tissues [13]. However, the connection between IFF and drug distribution within the brain is severely understudied [14]. CED emerged as a technique to overcome the poor penetration of the tumor and increase therapeutic distribution at the lesion site by creating pressure differentials to increase convective fluid flow directly to the tumor and its surrounding area [12]. Despite being a promising proposal, CED has thus far failed to show significant improvements for patients in clinical trials. One limitation of CED is that infusate can escape the targeted area by following natural flow trajectories within the brain [15]. This observation has reinforced the notion that drug delivery is linked to natural and pathological flow patterns of the brain, which are not always predictable. It is generally thought the increased tumoral pressure drives IFF out of the tumor into the interstitial space, with the highest flow velocities at the edge of the tumor bulk [16,17]. This phenomenon has been shown in implanted preclinical tumors and in some patient tumors outside of the brain [17,18]. Though this flow pattern is definitely observed, there are also areas with inward flow, parallel flow, and no flow seen at the boundary of a single tumor in implanted murine models [19]. Thus, there is accumulating evidence that although flow patterns are undoubtedly affected by the heightened pressure of the tumor bulk, this does not lead to a single, uniform flow pattern, either intra- or intertumorally. Hence, having a solid understanding of the mass transport mechanisms, including convection and diffusion, at and around the lesion site is critical to develop effective solutions for the longstanding obstacles in drug delivery. 

Fittingly, methods to measure and model these parameters have been a source of growth in the past few decades, with advances in both imaging via MRI and computational approaches. Diffusion-based imaging techniques, such as diffusion tensor imaging, offer insight into the transport of small molecules throughout the central nervous system [20]. Such advancements in imaging techniques have led to the use of MRI as a tool for estimating drug distribution within the brain. This has been done by creating MRI-visible drug delivery systems or by correlating specific imaging parameters with drug concentrations at known locations [21,22]. In this study, our goal was to focus on IFF imaging to give further data to computational modelers and drug delivery experts to gain a better understanding of what transport looks like within and around human GBM. Dynamic contrast-enhanced magnetic resonance imaging (DCE-MRI), which utilizes a paramagnetic contrast agent such as gadolinium, is a well-suited imaging modality to analyze IFF because it allows for the quantitative and noninvasive determination of parameters such as tissue diffusivity and transport within brain tissue [23]. Thus, we used DCE-MRI to study the transport of flow and therapeutics within and around tumors.

## 2. Materials and Methods

### 2.1. The Cancer Imaging Archive Ivy GAP Database

The Ivy Glioblastoma Atlas Project (Ivy GAP) database was accessed between July 2019 and July 2020 to select GBM patients from the 42 total archived pieces of patient information. The patient data were only analyzed if an axial, T1-weighted DCE-MRI that was devoid of motion artifacts was available [24,25]. Thus, we analyzed 14 of 42 patients from the Ivy GAP database: W13, W18, W29, W30, W31, W33, W34, W35, W36, W38, W40, W43, W48, and W50. Eight of these patients—W13, W33, W34, W35, W36, W38, W43, and W48—had pre- and post-resection DCE-MRI available. The sizes of the aforementioned groups were based on DCE-MRI availability in the Ivy GAP database rather than statistical sample size calculations. Larger datasets may be needed in further studies regarding IFF and transport in GBM. All data are publicly accessible via The Cancer Imaging Archive (TCIA) [25]. Additionally, Appendix A contains clinical information from the database color coded to match graphs for the patients who were analyzed in this study.

### 2.2. Convection and Diffusion Analysis

The analysis of IFF in the DCE-MRI acquired from the Ivy GAP database was performed using a computational model previously developed by our group [19]. Assuming that the MR signal intensity is proportional to the contrast concentration within the tissue allows the model to evaluate the spatiotemporal evolution of the contrast agent. This model requires an input of an image stack consisting of at least one pre-contrast (necessary for background subtraction from post-contrast images) and at least three post-contrast images of a single slice (which includes the tumor) from the full brain scan. The graphical user interface (GUI) associated with the model is used to draw a polygon around the region of interest (ROI) (i.e., the tumor) on the image and specify the resolution of the image, timing between the slices of the stack, etc. The model uses the image stack and information input in the GUI to calculate the isotropic diffusion coefficient and velocity field of an ROI by solving the diffusion–advection partial differential equation (PDE) below using the forward-time, central-space finite difference method.
(1)$$\frac{\partial \varphi \left(x,t\right)}{\partial t}=\;\nabla \cdot \left[D\left(x,t\right)\nabla \varphi \left(x,t\right)\right]-\nabla \cdot \left[\varphi \left(x,t\right)u\left(x,t\right)\right]$$

In the above equation, the contrast concentration given by *φ*(***x***,*t*), the isotropic diffusion coefficient given by *D*(***x***,*t*), and the velocity field given by ***u***(***x***,*t*) evolve in space (***x*** = (*x*, *y*)) and time (*t*). The details regarding the solutions of the above PDE and the model can be found in our previous publication [19,26]. Using estimates of the spatio-temporal evolution of the contrast agent as input, the model allowed us to infer the spatially-resolved diffusion coefficient and the vector field of IFF velocity. The mean and median values of the flow parameters of several slices per tumor were averaged to calculate overall parameter values for the entire tumor. These averages and vector fields were used for the various methods of data visualization presented here.

### 2.3. Statistics and Graphing and Generation of Figures

Statistical analyses were conducted on individual datasets as described in the results. Graphs were generated using GraphPad Prism v9.0 (GraphPad Software Inc., San Diego, CA, USA), and graphics were generated using Biorender (BioRender, Toronto, ON, Canada), a web based illustration tool, with a license to the corresponding author. The rose plots were generated using a modified version of the wind rose code on MATLAB R2020a (MathWorks, Natick, MA, USA), downloaded from the MathWorks File Exchange and created by Daniel Pereira. The heat maps and images with streamlines were generated using a Python script generated by our group.

## 3. Results

### 3.1. Interstitial Flow and Diffusion Coefficients Can Be Calculated from DCE-MRI

As we earlier demonstrated in mice, we were able to use gadolinium transport to simultaneously model both interstitial fluid velocity and diffusion coefficient from four sequential images. In mice, a specific sequence was required that took four images after gadolinium entry into the interstitial space over 12 min (one image every three minutes) [19]. In The Cancer Imaging Archive patient data were available for DCE-MRI image acquisitions, which took approximately 1200 images over the course of two-to-three minutes. We chose to analyze a set of images spanning the imaging session from these data and were able to successfully execute our analysis similarly to in mice to determine both IFF and diffusion coefficients in and around the tumors. Our overall process is shown in Figure 1 and includes the acquisition of images from the database, followed by use of the Lymph4D analysis tool [19,26] to generate the data in a pixel-wise fashion, and then subsequent data visualization using MATLAB and Python.

### 3.2. Interstitial Fluid Flow Magnitude Is Variable across Patients

We analyzed both flow within the tumor and within the surrounding parenchyma, with the hypothesis that velocity would be faster in the parenchyma than within the tumor (Figure 2A). Six MRI slices per patient were analyzed, which encompassed the majority of the tumor in each patient. The average velocity magnitude was calculated per slice, and then these were averaged to comprise a total mean velocity magnitude on a per patient basis. Generally, there was about a 10–20% range of mean tumor velocities among the six slices that was not inherently dependent on location within the brain (Appendix A).

We detected no significant difference between the velocity magnitude as measured within the tumor as compared to the surrounding parenchymal space (Figure 2B). Furthermore, we found that the rate of flow within the tumor significantly and strongly correlated with that of the parenchyma (Figure 2C), thus indicating that the patient-by-patient variability may be more important than the macroregional differences within these individual tumors. To compare, we performed the same analysis on diffusion coefficients. We did not detect a significant difference in the diffusion coefficient as calculated in the two regions across patients (Figure 2D), but we again found a significant, though moderate, correlation between the calculated diffusion within the tumor compared to the surrounding space (Figure 2E). We did not observe that the size of the tumor correlated with the velocity (Appendix A). This was similar to a lack of correlation previously observed in mice.

### 3.3. Patient Survival Correlates Positively with Mean Velocity Magnitude

We aimed to examine the effect of patient-specific variables on flow velocity magnitude within the dataset. We found that the correlation between velocity magnitude and patient weight was nonexistent (r = −0.0055 and *p* = 0.984) (Appendix A). We did not find that there was a significant difference between sexes (*p* = 0.147), methyl guanine methyl transferase (MGMT) methylation status (*p* = 0.9497), or epidermal growth factor receptor (EGFR) amplification (*p* = 0.329), though this dataset may have been slightly too underpowered to conclude that there is no effect (Appendix A). Interestingly, we did find that age significantly correlated with a lower IFF velocity magnitude throughout the tumor (Figure 3A). This may be explained by a host of literature indicating that fluid flow within the brain slows with age, as documented by the MRI of ventricles, blood vasculature, and drainage pathways. Most importantly to clinical outcomes, we found that the mean velocity magnitude within the tumor significantly correlated with survival, with higher rates of IFF velocity correlating with longer survival times (Figure 3B). As expected, age correlated negatively and significantly with survival as well (Figure 3C) [27,28]. However, to firmly conclude the correlation between velocity and survival, without age as a potential confound, a larger dataset with a range restriction on age would be valuable to examine this novel interaction. Contrastingly, we did not find any correlation between diffusion coefficient and survival in this patient cohort (r = 0.182; *p* > 0.05, not significant, Appendix A).

### 3.4. Resection of Tumor Does Not Eliminate Interstitial Fluid Flow

It appeared that the inherent velocity was a patient-specific parameter more than an interpatient parameter based on our tumor and parenchymal analysis. Thus, we aimed to examine the effect of resection on interstitial velocity magnitude. There was a subset of eight patients in the TCIA Ivy GAP database for which pre- and post-resection DCE-MRIs were available (Figure 4A). Analyzing these patients revealed that there was not a significant decrease nor an increase in interstitial velocity magnitude pre- and post-resection across our cohort (Figure 4B). However, we did see that there was a change in velocity for individual patients that could be physiologically relevant for better understanding treatment post-resection, with some patients showing decreased flow vs increased flow. Potentially due to this variability in patient response post-resection, we did not see a significant correlation between pre- vs. post-resection interstitial velocity magnitude (Figure 4C). Thus, though there is still inherent flow in the parenchymal space post-resection, the effects of changes in this velocity are unknown within this cohort.

### 3.5. Directional Flow Velocity is Patient-Specific

Analyzing IFF in numerous patients revealed an inherent variability in IFF directionality within GBM. Some patients were found to have a relatively uniform IFF, with little preference for a specific direction, whereas others were found to have a strong tendency to flow in a particular direction. The contrast between these flow patterns was made with analyses for patients W43 (Figure 5A–D) and W48 (Figure 5E–H). The post-gadolinium T1-weighted images of patient W43 (Figure 5A) and patient W48 (Figure 5E) indicated the location of these patients’ tumors. The streamlines for patient W43 were heavily oriented towards the anterior brain (Figure 5B), whereas no such clear distinction could be made for the streamlines of patient W48 (Figure 5F). Furthermore, examining the quiver plot in addition to the streamlines seemed to indicate that the areas of faster flow and more directional flow correlated for both patients. This could also be visualized by examining the velocity magnitude heat maps for patient W43 (Figure 5C) and W48 (Figure 5G) in conjunction with their corresponding streamlines. Conversely, since faster flows were also observed in the nondominant flow directions, the rose plot of velocity magnitude and direction challenged the notion that IFF magnitude and direction are always correlated (Figure 5D). Finally, the major advantage of the rose plots is that they effectively represented the directionality vs. uniformity of IFF in patient W43 (Figure 5D) vs. W48 (Figure 5H). Though the underlying reasons and accompanying effects of tumoral flow patterns may still be up for debate, we showed both intra- and intertumoral heterogeneity in IFF patterns.

## 4. Discussion

Here, we have described the use of a previously developed technique to examine IFF within and around the patient GBM microenvironment. One of the most interesting findings of this study was that interstitial fluid velocity magnitude correlated with survival in this patient cohort, such that an increased survival was associated with an increased convective flow velocity. We did not see a similar trend with changes in fluid diffusion. This suggested that convective transport may be more important in GBM prognosis in some regard. A possible explanation for the correlation between flow velocity and survival is improved drug transport within the tumor towards the parenchyma. We note that transport into and within the tumor should be distinguished. Since we did not examine parameters related to transport into the tumor (such as K-trans or vessel permeability), we do not know if these values would potentially correlate with our measurements of velocity [29,30]. Such correlations would indicate whether velocity is simply indicative of increased delivery into the tumor and not necessarily through the tumor into the parenchyma. Higher interstitial pressures that limit transport across the vasculature are expected to lead to reduced transport into the tumor [31,32]. Efforts to increase pressure driving flow from vessels into tumors have shown preclinical success [32,33]. Generally, higher interstitial fluid pressures within the tumor are indicative of higher interstitial fluid flow in the tumor periphery as evidenced in multiple preclinical solid tumor models [15,17,31,34]. Thus, it is not readily apparent from our data that transport into the tumor and transport through the tumor correlated in this dataset. However, this new information regarding the correlation of survival and convective flow may be indicative of the benefits of enhanced drug delivery in GBM.

Since interstitial pressure is altered with resection of the tumor bulk, we examined the effect of surgery on the transport and IFF around the resection cavity. The resection of the tumor reduces interstitial pressure within the cranium and tumor surroundings by alleviating the source of increased pressure from the brain [35,36]. Thus, it was expected that the removal of the tumor bulk would result in a reduced IFF velocity [37]. Contrary to this expectation, we found that the velocities across patients pre- and post-surgery were generally similar, with some cases of reduced or heightened velocities seen on an individual patient basis. Thus, whether the tumor bulk is intact or not, IFF is still occurring within the surrounding tissue. However, one caveat is that we do not know the normal IFF rate in a healthy, non-tumor bearing brain. Thus, it is unclear if patients who had relatively lower velocities showed increased velocity beyond normal levels as compared to patients with heightened velocities. Indeed, measurements of cerebrospinal, perivascular, and vascular flows indicate a range of patient-to-patient variability in flows [38]. Some of this variability in flow may be related to disease and/or physiological characteristics, whereas some has been linked to age [39,40,41]. The negative correlation we saw between age and interstitial fluid transport is important since we also saw a negative correlation between flow velocity magnitudes and survival. Preclinical and clinical studies of therapeutics in patient-specific models that incorporate parameters such as age, sex, and other clinical characteristics in relation to flow characteristics are warranted.

Though flow velocity magnitudes can provide insight into transport within the tumor, the direction of flow was also an important parameter explored in this study. As expected, our analysis of flow directionality yielded interpatient variability in flow patterns in and around the lesion. Some tumors exhibited a uniform flow, whereas others exhibited flows with a dominant spatial direction. Visualizing the velocity magnitude and direction concurrently (Figure 5) showed that, somewhat surprisingly, the fastest flows were not always in the dominant direction of flow. The mechanisms determining the observed flow patterns were difficult to decipher with our analysis method. However, we suspect that flow patterns are the result of a combination of factors such as natural flow pathways within the brain, changes in the transport properties related to the extracellular matrix of the tumor and its microenvironment, and the increase or decrease of available fluid by way of the tumor modulating the surrounding circulation and immune response [42,43,44] 

Regardless of the underlying mechanisms, IFF flow patterns will likely have significant implications because drug delivery is intrinsically linked to the diffusion and convection within a tissue [45]. Convective forces dominate when therapeutics are larger (i.e., antibodies and nanoparticles) compared to smaller (i.e., small molecules and peptides). IFF allows us to specifically look at the advection component of transport, which is important for determining the trajectory of therapeutic transport within the tissue. In addition to having important implications for drug delivery, IFF patterns may have connections to disease progression. For instance, previous preclinical studies have indicated that there are mechanisms by which flow mediates increased invasion [46,47]. Thus, it is possible that areas of fast and/or directional flow may be at greater risk for invasion and consequent recurrence.

While the application of our technique map IFF is novel and gave us insight into the transport parameters of a patient’s unique tumor, there are limitations to consider. We do not have the 3D resolution of the tumor nor transport parameters at this point because our current analyses were performed on 2D planes. This was because 3D flow, represented as vectors in the x–y plane, may obfuscate transport in the z direction. Thus, any holistic approach to applying these data for overall transport modeling is limited. This may be particularly apparent when examining inter-slice variability, where some patients exhibit a wider range of flow magnitudes depending on the slice through the tumor. We plan to improve our approach and address these limitations in future studies.

Overall, IFF patterns are a promising avenue to explore while determining patient-specific reasons for therapeutic failure and disease progression. However, we do not see IFF analysis as a “magic bullet” for GBM research or treatment. Coupling our method of patient-specific IFF mapping with more advanced mathematical modeling approaches related to predicting therapeutic response and disease progression will offer further insight into the promising field of personalized mathematical oncology [48,49,50].

## 5. Conclusions

The overarching goal of this study was to use our previously developed computational model along with DCE-MRI from the Ivy GAP database to study IFF and mass transport mechanisms in human patients with glioblastoma. Interesting findings of this study include the significant correlation between survival and age with mean tumor velocity magnitude, the presence of IFF despite the surgical removal of the tumor bulk, and the inter-tumoral variability in the directionality and overall patterns of IFF. Our findings indicate that there are likely significant connections between clinical outcomes in GBM, IFF, and convective therapeutic transport. However, further studies need to be performed to better understand these connections and to integrate IFF with the broader field of personalized mathematical oncology.

## Figures and Tables

**Figure 1 pharmaceutics-13-00212-f001:**
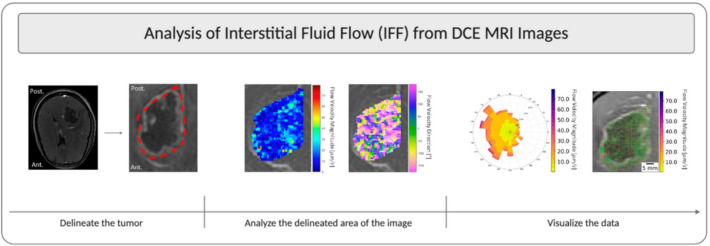
Overview schematic of IFF analysis using DCE-MRI. This figure illustrates the steps for analysis using patient W13 from the Ivy Glioblastoma Atlas Project (GAP) database as an example. The tumor was located on a slice of interest, all timepoints of the slice of interest were extracted from all DCE acquisitions, and then the tumor was delineated and analyzed using the Lymph4D analysis tool. The component-wise velocity vectors from the analysis were then input into other MATLAB R2020a and Python 3.6 scripts to develop images overlayed with a streamline and a quiver plot, as well as the rose plots of velocity magnitude and direction.

**Figure 2 pharmaceutics-13-00212-f002:**
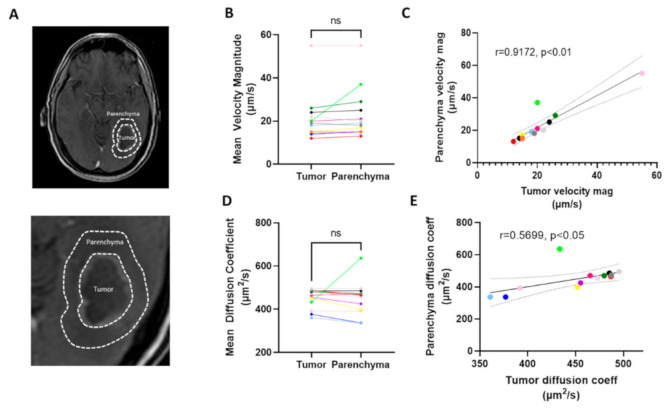
Transport parameters measured in the tumor vs. the surrounding parenchymal space. (**A**) Representative images of tumor (T) and parenchymal space (P) highlighted in patient W-36. (**B**) Paired data from individual patients. (**C**) Correlation of parenchymal vs tumor velocity magnitude averaged over six slices. (**D**) Mean diffusion coefficient for same regions averaged over six slices. (**E**) Correlation of diffusion coefficient by patient. Correlations are shown with a 95% CI, and colors indicate different patients and are consistent across Figure B–E. Each color represents a unique patient (see also Appendix A), ns = not significant, CI = confidence interval.

**Figure 3 pharmaceutics-13-00212-f003:**
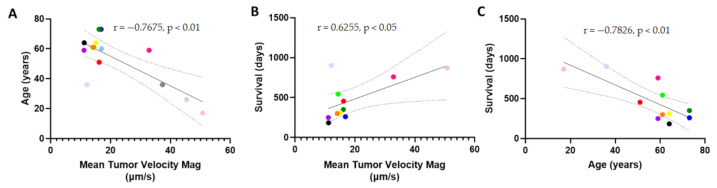
Patient survival and age correlate with tumor interstitial fluid velocity magnitude. (**A**) Age vs. mean tumor velocity magnitude. (**B**) Survival in days vs the mean tumor velocity magnitude averaged over six slices per patient. (**C**) Age vs. survival in patients analyzed for interstitial fluid velocity. Correlations are shown with 95% CI, and colors indicate different patients and are consistent with Figure 2, CI = confidence interval.

**Figure 4 pharmaceutics-13-00212-f004:**
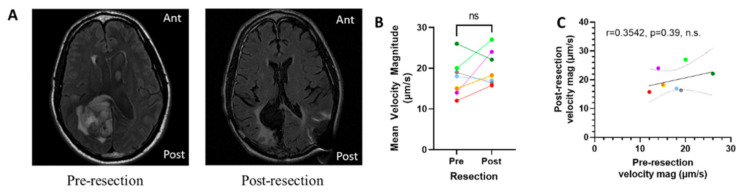
Interstitial fluid velocity is apparent after tumor resection. (**A**) Representative images from a pre- (left) and post- (right) resection tumor (Patient W-33). (**B**) Change in mean velocity magnitude by patient pre- and post-resection. (**C**) Correlation of pre-resection velocity with post-resection velocity magnitude. Correlations are shown with 95% CI, and colors indicate different patients and are consistent with Figure 2 and Figure 3, ns = not significant, CI = confidence interval.

**Figure 5 pharmaceutics-13-00212-f005:**
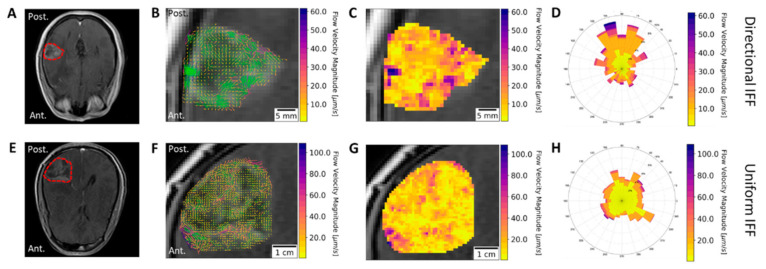
Directional versus non-directional IFF in glioblastoma (GBM) patients. Post gadolinium T1-weighted images of patients (**A**) W-43 and (**E**) W-48, with tumors outlined. Results from IFF analysis consist of (**B**,**F**) images including streamlines (green) and vectors (color bar) for velocity, (**C**,**G**) heat maps, and (**D**,**H**) rose plots. The results from patients W43 and W48 are plotted as examples of directional and uniform IFF, respectively.

## Data Availability

Data was acquired through publicly accessible database, The Cancer Imaging Archive.

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
