# Peer review of "Utilizing Dynamic Contrast-Enhanced Magnetic Resonance Imaging (DCE-MRI) to Analyze Interstitial Fluid Flow and Transport in Glioblastoma and the Surrounding Parenchyma in Human Patients"

_pharmaceutics, 2021, doi:10.3390/pharmaceutics13020212_

Round 1
Reviewer 1 Report
In this manuscript, extensive revisions are required and several concerns need to be addressed as follows:
1. The title should be representative of the study. The aim of the study is the use of dynamic contrast-enhanced magnetic resonance imaging (DCE-MRI) to study the transport of flow and therapeutics within and around glioblastoma. However, the authors have ignored DCE-MRI despite being the main element of the study. In addition, "in patients" should be completed.
2. The abstract has been written in a very confusing way. The authors used a long introduction, very concise methods, and non-organized methods. The abstract needs to be rewritten more informative and organized way.
3. Keywords: replace "DCE-MRI" with "Dynamic contrast-enhanced magnetic resonance imaging".
4. There is a problem in using abbreviations throughout the manuscript especially the abstract. The full term should be mentioned first with the abbreviation between paresis then the abbreviations should be used throughout the manuscript. E.g. line 21: MRI should be mentioned first as magnetic resonance imaging (MRI) then the abbreviation "MRI" used further.
5. The results section is a serious drawback of the current study. This section in its present form is a mix of the introduction, methods, and discussion sections as follows:
- The first subtitle should be transferred to the methods section.
- Lines 153-156: the authors explain their earlier study in mice. Where is the reference for this method? Also, why this illustration present in the result section.
- Lines 177-181: this information should be either present in the introduction or discussion section.
- Lines 237-241 should be transferred to the discussion.
6. The discussion section needs to be reframed after revising the introduction.
7. The conclusion section is missed.
8. The manuscript contains many typing errors, grammatical, formatting, and styling errors (e.g. Lines 190-200: We saw, We did see, We saw, We did not observe). Thus, the manuscript needs to be carefully revised for the English language and formatting by a native English speaker.
Author Response
We thank the reviewer for their careful read of our manuscript and their comments. We have responded to each comment below and incorporated those changes into the manuscript.
In this manuscript, extensive revisions are required and several concerns need to be addressed as follows:
- The title should be representative of the study. The aim of the study is the use of dynamic contrast-enhanced magnetic resonance imaging (DCE-MRI) to study the transport of flow and therapeutics within and around glioblastoma. However, the authors have ignored DCE-MRI despite being the main element of the study. In addition, "in patients" should be completed.
Per this comment, the title of the manuscript was updated as follows: “Utilizing Dynamic Contrast Enhanced (DCE)-MRI to analyze interstitial fluid flow and transport in glioblastoma and the surrounding parenchyma in human patients.”
- The abstract has been written in a very confusing way. The authors used a long introduction, very concise methods, and non-organized methods. The abstract needs to be rewritten more informative and organized way.
The abstract was revised to have a slightly shorter introduction and slightly lengthier methods. Additionally, the overall abstract was revised to be a clearer summary of the study. The general format of the abstract was dictated by the journal guidelines. We thank the reviewer for this comment and hope it is clearer now.
- Keywords: replace "DCE-MRI" with "Dynamic contrast-enhanced magnetic resonance imaging".
We thank the reviewer for this suggestion. We have included "Dynamic contrast-enhanced magnetic resonance imaging” and the abbreviation “DCE-MRI." We believe that both of these terms may be searchable.
- There is a problem in using abbreviations throughout the manuscript especially the abstract. The full term should be mentioned first with the abbreviation between paresis then the abbreviations should be used throughout the manuscript. E.g. line 21: MRI should be mentioned first as magnetic resonance imaging (MRI) then the abbreviation "MRI" used further.
We thank the reviewer for this comment as it will make it easier to read through the manuscript. The abbreviations used in the manuscript were updated to follow the guidelines in this comment.
- The results section is a serious drawback of the current study. This section in its present form is a mix of the introduction, methods, and discussion sections as follows:
- The first subtitle should be transferred to the methods section.
As the reviewer mentioned, the first subtitle could also be placed in the methods section. However, we chose to place it in the results section because we would like to highlight that this is a novel application of this methodology. This is our first publication showing that this analysis can be performed using human patient data.
- Lines 153-156: the authors explain their earlier study in mice. Where is the reference for this method? Also, why this illustration present in the result section.
We thank the reviewer for catching that we missed the reference for that work. We have added it to the manuscript. The illustration placement was a result of the formatted document used per the journal specifications.
- Lines 177-181: this information should be either present in the introduction or discussion section.
We thank the reviewer for this comment. We agree that this information is better suited for the introduction. Thus, it was moved to the third paragraph of the introduction.
-Lines 237-241 should be transferred to the discussion. These lines have been moved to the discussion.
- The discussion section needs to be reframed after revising the introduction.
We have gone through the discussion in the context of the new introduction and have edited it throughout. If there are more specific suggestions of content or points to amend or expand upon, we would be happy to incorporate them.
- The conclusion section is missed.
We thank the reviewer for this comment. Though the journal does nor require a conclusion for shorter manuscripts, we have added a brief conclusion for clarity.
- The manuscript contains many typing errors, grammatical, formatting, and styling errors (e.g. Lines 190-200: We saw, We did see, We saw, We did not observe). Thus, the manuscript needs to be carefully revised for the English language and formatting by a native English speaker.
Per this comment, the wording of some parts found within lines 190-200 have been altered to give the paper a better flow. Additionally, the corresponding author and several other authors on this manuscript are native English speakers, and have reviewed the manuscript for grammar and typing errors. If the concern is the style of the writing, we prefer to write in an active voice instead of passive voice but this is not incorrect.
Reviewer 2 Report
The authors have presented a very interesting exploratory article regarding the impact of fluid flow in the setting of the cerebrovasculature. Specifically, the authors highlight the difficulty in treating glioblastoma tumours given their pathophysiology, but also hone in on the therapeutic potential in the increase in vascular permeability seen at the site of the glioblastoma tumours. In that way, drug delivery at the site of the tumour should be more efficient, given the restrictive nature of the blood brain barrier, and while many papers have focussed on mechanisms of delivery, it is refreshing to see a paper focus on the hemodynamic parameters which confound same efforts. Overall, this was a very interesting read.
In reading the manuscript I have a number of suggestions. The authors should address/consider the following when preparing a suitable revision.
- In the methods the authors mention that they selected 14 patients from an existing data set, of which 8 had information pre- and post-resection. Was this the maximum number of participants available, or was this number determined by some sort of power calculation? Could more participants have been added to the analysis, or is the number selected justified?
- It would be useful if certain information were made available/included in this article. For example, more information on each of the patients included in the analysis, or details on the study on which this is based. Often these archives are restricted access, and it would be useful if more details on those and similar were included, if even as supplementary.
- It might be worth considering colour coding the data for each patient such that trends across the study could be correlated.
Author Response
We thank the reviewer for their thoughtful comments regarding our submission. We have addressed the comments below and within the revised submitted manuscript.
The authors have presented a very interesting exploratory article regarding the impact of fluid flow in the setting of the cerebrovasculature. Specifically, the authors highlight the difficulty in treating glioblastoma tumours given their pathophysiology, but also hone in on the therapeutic potential in the increase in vascular permeability seen at the site of the glioblastoma tumours. In that way, drug delivery at the site of the tumour should be more efficient, given the restrictive nature of the blood brain barrier, and while many papers have focussed on mechanisms of delivery, it is refreshing to see a paper focus on the hemodynamic parameters which confound same efforts. Overall, this was a very interesting read.
In reading the manuscript I have a number of suggestions. The authors should address/consider the following when preparing a suitable revision.
- In the methods the authors mention that they selected 14 patients from an existing data set, of which 8 had information pre- and post-resection. Was this the maximum number of participants available, or was this number determined by some sort of power calculation? Could more participants have been added to the analysis, or is the number selected justified?
This is the maximum number of participants available in the database that could be used in our study. Selection from the database is based on the patients having glioblastoma diagnosis and DCE-MRI availability. The manuscript is edited to state this more explicitly under the section “2.1 The Cancer Imaging Archive Ivy GAP Database.” We thank the reviewer for this comment as it allowed us to clear up potential confusion for future readers.
2. It would be useful if certain information were made available/included in this article. For example, more information on each of the patients included in the analysis, or details on the study on which this is based. Often these archives are restricted access, and it would be useful if more details on those and similar were included, if even as supplementary.
We thank the reviewer for this insightful comment. We have created a supplementary table (Table S1) including more detailed information regarding the patient and their disease. Furthermore, there is a large amount of data available for these patients within the databases which we have made sure to both link to and cite within the manuscript so readers can search at their leisure.
3. It might be worth considering colour coding the data for each patient such that trends across the study could be correlated.
Thank you for this great suggestion, the data has all been color-coded by patient in all graphs throughout the manuscript. Further, we have included these color-codings in the now included table in the supplement for even more clarity. We thank the reviewer for the suggestion as it both makes the data more transparent and colorful.
Round 2
Reviewer 1 Report
-